# Permeability and Load Capacity of Iron Porous Bearings with the Addition of Hexagonal Boron Nitride

**DOI:** 10.3390/ma15155112

**Published:** 2022-07-22

**Authors:** Krzysztof Gocman, Tadeusz Kałdoński, Bolesław Giemza, Artur Król

**Affiliations:** 1Faculty of Mechanical Engineering, Military University of Technology, 00-908 Warsaw, Poland; tadeusz.kaldonski@wat.edu.pl; 2Propellants and Lubricants Division, The Air Force Institute of Technology, Ksiecia Boleslawa St. 6, 01-494 Warsaw, Poland; boleslaw.giemza@itwl.pl; 3Institute of Vehicles and Transportation, NATO Defense College, Via Giorgio Pelosi 1, 00143 Rome, Italy; a.krol@ndc.nato.int

**Keywords:** self-lubrication, friction, porous metal bearings, iron sinter, boron nitride

## Abstract

Due to their properties, porous sliding bearings are considered to be maintenance-free, which means that no lubrication is required during operation. Their design enables operation at low rotational speeds with high load. Another effect of this bearing design is the lubrication continuity in the tribological pair. In this study, the selected tribological properties (such as load capacity and permeability) of new-generation iron porous bearings with boron nitride powder were experimentally investigated. Tests were carried out under various conditions, using unique test apparatus. The addition of 3% (weight) of hexagonal boron nitride (hBN) significantly increased the load capacity of tested porous bearings in comparison to the same standard bearings containing 2.5% copper. The obtained p_gr_·v rating reached almost 7 MPa, which is a value almost three times higher than the requirements for this type of bearing. It is worth emphasizing that such a result was obtained despite a noticeable deterioration in the air and oil permeability of the bearing.

## 1. Introduction

Porous slide bearings are now widely used in industrial equipment as well as in off-road trucks, military combat vehicles, tractors, heavy engineering machines, and other technical devices. The field of application of porous slide bearings is constantly expanding. In the automotive industries of highly developed countries, sintered bearings are widely used, sometimes replacing rolling bearings. Sintered bearings are self-lubricating and therefore do not require labor-intensive maintenance. Porous sleeves are widely used in devices where conventional lubrication is ineffective (e.g., at low temperatures, in a vacuum, in a chemically active environment, etc.), in machines and devices where conventional lubrication may cause contamination of the product (e.g., in the textile, food, paper industry, etc.), and in cases where lubrication service is difficult or even impossible.

Porous iron-based slide bearings are being produced as pure iron bearings without any additions, with a small graphite addition (0.4–1.0% or 0.5–2% of C), and with considerable copper content (1–5% or even 15–25% Cu) with or without graphite. There are also iron bearings being made with up to 3.5% lead added, depending on the application and needs, as well as iron bearings with added manganese, silicon, tin, nickel, or molybdenum disulfide [1,2,3,4,5,6,7,8,9].

Iron sleeves (with a density of about 5.6–6.0 g/cm^3^) are being used in medium-loaded machine elements. The addition of graphite improves the noiseless operation of the bearings and increases the lubricant film strength. This makes it possible to operate with a larger bearing clearance in comparison to that required for non-graphite bearings. However, excessive graphite content (over 3%) worsens the mechanical properties of the sleeve. Iron sleeves with added copper were introduced to the industry mainly due to their increased endurance properties. Compared to other types of porous slide bearings, these bearings can be more heavily loaded and can be used with a higher sliding speed. Copper is being added to iron bearings in quantities up to 25%, which results in a significant increase in hardness and resistance to impact loading. Commonly used porous sleeves designed for work with moderate load and speed usually contain about 2.5% Cu (for example, the reference sleeves used in this research, marked as T-1-x). The porous slide bearings with additional lead have lower durability at higher speed than the standard iron bearings, but they demonstrate better properties at lower speed and higher loads.

On the other hand, sleeves sintered from iron powder with the addition of a selected amount of hexagonal boron nitride (h-BN) to allow for increased load-bearing capacity and durability, and with the use of appropriately selected lubricating oil, are not widely known, at this time [2,6]. In this study, the maximum content of boron nitride in the tested bearings was set at 3% by weight, which resulted from previous tests carried out by the authors of [10,11,12,13,14,15]. In those studies, it was found that porous samples containing up to 3–5% of h-BN in the sinter [10,11,12,13,14,15] were generating a friction coefficient several times lower than that for porous samples without h-BN (when lubricated, e.g., with gear oil). The research in [12,13,14,15] shows that the 5% content of hexagonal boron nitride in the iron sintered material is an acceptable limit. Taking into account all the results related to these tests (hardness, durability, wear resistance, thermal conductivity, frictional resistance), it was found that 3% content of boron nitride is the most advantageous, and the decision was made to produce such porous sleeves and designate them for further research.

A critical parameter determining the functional properties of porous bearings is their air and oil permeability, defined according to the appropriate liquid used for impregnation/lubrication of the sleeve. The key issue is to assess the variability of air/oil permeability over time. This variability is largely due to the nature of the chemical and physical interactions between the porous structure of the bearing and the lubricating oil.

In this paper, we attempt to assess the influence of the presence of boron nitride as a sinter component on the functional properties of porous sleeves. For this purpose, their permeability and load-bearing capacity have been tested.

## 2. Experimental Details

Following the results of the preliminary test [10,12,13,14,15], a series of porous sleeves, made of iron sinter containing up to 3% h-BN (marked as T-3-x) were produced.

The basic material for the production of the porous sinters was an iron powder manufactured by the Swedish company Höganäs AB, with the symbol NC 100.24. The sleeves used in the research were produced using two-sided pressing, during which the powder was in the matrix between two stamps. To increase the slip of the stamps, the matrix used for stamping was covered with a thin layer of graphite powder. The obtained samples were subjected to sintering and calibration, during which the sleeves obtained the desired dimensions with the required tolerance and the appropriate surface roughness. As a result of this process, samples with an open porosity within 20% (for samples without boron nitride) and approx. 17% (for samples with the addition of hBN) were obtained. Open porosity was determined by measuring weight gain when the product was impregnated with oil, taking into account the apparent and theoretical sinter density and the oil density [12,14].

The standard T-1-x (containing 2.5% of Cu) and a new-generation T-3-x (containing h-BN) porous sleeves were tested during the comparative experiments on the load capacity (load-carrying ability) and permeability. The letter “x” represents a unique, sequential number of the tested porous sleeve. The tested porous sleeves had an inner diameter of 25 mm, an outer diameter of 35 mm, and a length of 20 mm.

### 2.1. Permeability

The permeability of the sleeve depends on the size of the pores and their distribution, so it is influenced by the grain size distribution of the powders and the shape of the grains. It determines the flow resistance and the circulation of the lubricant in the bearing. The amount of the lubricant and the conditions of its circulation between the lubrication gap and the porous structure have a significant influence on the operation of a self-lubricating bearing. The permeability of the sleeve is therefore a very important parameter on which the operational properties of the porous bearing depend, including durability and load capacity [14]. Standardization documents for permeability determination recommend the use of gases due to the phenomenon of the formation of a layer of surface-active compounds (additives) inside the pores, as described in the literature [16]. However, the purpose of this measurement was also to determine the actual permeability for the given oil with which the bearing was impregnated. The oil circulation in the porous sleeve and the lubricating gap results from the permeability; therefore, the value of this parameter and its changes over time determine the bearing capacity and durability. Consequently, the permeability was assessed for both air and lubricating oils.

The air permeability of the sleeve was assessed in accordance with [16]. The study relies on determining the volumetric airflow rate and the pressure drop of air of known viscosity while passing through a Ø25/Ø35 × 20 mm (sleeve dimension) porous sample of known active surface and thickness, under the laminar flow conditions, on the test stand, as shown in Figure 1. The oil permeability of the sleeve for actual lubricants was determined according to the same standard as for the appropriately prepared test stand. The permeability was measured after closing the sleeve axially between the two planar surfaces. Air/oil was passed through the sleeve wall to the outside. The face surfaces of the tested sleeves were separated from the flat surfaces by a pair of rubber gaskets to overcome the unevenness of the surfaces, to prevent leakage of the fluid. A Vegabar 14 differential pressure sensor with 1% accuracy was used to determine the pressure. A Pt100 thermocouple with an accuracy of 0.1 °C was used to determine the fluid temperature. Measuring cylinders with a capacity of 10 cm^3^ were used to determine the volumetric flow rate.

### 2.2. Load Capacity

The tests were carried out using two different test stands: PLS-01 and PLS-02. Various oils were used as lubricants (for impregnation). Some of the most effective ones included: Hipol-15F 85W/90 (marked as 0-3), Mobilube 1SHC 75W/90 (marked as O-26), and the mixture of basic polyalphaolephin oils PAO-8+PAO-40 (55% + 45% by weight, respectively; marked as O-30). The other oils that were tested, such as Antykol TS120 (recommended by Polish producers) and PFPE oil Klüberalfa DH3-100 (recommended by Klüber Lubrication) were significantly worse [12,13,14,15].

Both test stands were equipped with NC6 steel shafts (Ø25, h7 tolerance) polished to Ra = 0.32 µm with a hardness of approx. 65 HRC.

The test stand is presented in Figure 2. The device enabled simultaneous testing of up to 16 self-lubricating radial plain bearings under constant or stepwise load conditions in the range of 100, …, 3200 N and rotational speed up to 1500 rpm. The drive consisted of a pair of electric motors with a power of 4 kW each. This stand design ensured the regulation and registration of the rotational speed and loads of the tested bearings. It was capable of working in both manual and automatic control modes.

The stand consisted of the following components: a drive and transmission unit, a set of research modules, the measuring and archiving unit, and a stand work-signaling unit. The PLS-01 tester was designed and self-made by the Military University of Technology (MUT) and was intended specifically to test self-lubricating porous bearings.

The limit pressure (p_gr_) tests were carried out (PLS-01) for the three rotational speeds, i.e., n_1_ = 600 rpm (v_1_ = 0.79 m·s^−1^), n_2_ = 1000 rpm (v_2_ = 1.31 m·s^−1^), and n_3_ = 1400 rpm (v_3_ = 1.83 m·s^−1^). When stabilization (and/or reduction) of the resistance to motion as well as the operating temperature of the bearing was noticed, the load applied to the friction pair was successively increased. The load (pressure) applied to the bearing was increased by 0.4 MPa in each subsequent step. After determining the pressure causing bearing seizure (p_z_), the previous, lower value of the pressure was assumed as the limit pressure (p_gr_). The parameters listed below were defined as the assumed criteria for bearing seizure:Rapid increase in resistance to motion, with the moment of friction M_t_ greater than 2 Nm, and coefficient of friction µ exceeding 0.3;Unstable work of the bearing, i.e., high variations of parameters’ values;Rapid increase in the temperature of the bearing up to T > 200, …, 220 °C.

The PLS-02 test stand (with the same friction pair as PLS-01) allows the tests to be carried out with fluency, linearly increased load (up to 17 N/s, max. load–3200 N), and adjustable rotational speed (even up to 6000 rpm). The PLS-02 tester, like PLS-01, was specially designed for sliding porous bearings testing.

The tests with linearly increasing load were carried out with the same rotational speeds as the ones conducted on the PLS-01 stand. An increase in load (Q) occurred until there was a rapid increase in moment of friction (dM/dt), indicating breakdown of the lubricating film. In these tests, it was assumed that the quantity of energy supplied to the tribological system per unit of time was approximately the same, despite various rotational speeds, i.e.:For *n* = 600 rpm (v = 0.79 m·s^−1^): ΔQ = 14 N·s^−1^, ΔQ·v = 11.06 W·s^−1^;For *n* = 1000 rpm (v = 1.31 m·s^−1^): ΔQ = 8.4 N·s^−1^, ΔQ·v = 11.00 W·s^−1^;For *n* = 1400 rpm (v = 1.83 m·s^−1^): ΔQ = 6.0 N·s^−1^, ΔQ·v = 10.98 W·s^−1^.

All the tests were performed at room temperature (18–20 °C) in an air-conditioned laboratory room.

## 3. Results and Discussion

### 3.1. Air and Oil Permeability

In Figure 3, Figure 4 and Figure 5 the summary characteristics of both types of porous sleeves’ air and oil permeability are presented. Two selected oils: Hipol 15F 85 W/90 and Mobilube 1SHC 75 W/90 were used to measure the oil permeability of porous sleeves.

The results of the tests showed that the air permeability of T-3-x sleeves made of iron sinter containing hexagonal boron nitride was significantly lower than that for T-1-x standard sleeves made of Fe + 2.5% Cu sinter. Observed differences in air permeability arose because of the porosity differences, which were lower by about 3–4% for the sleeves containing h-BN in the sinter. As a consequence, the final average values of permeability of the sleeves with h-BN (T-3-x) for Hipol 15F oil were two times lower than for the standard sleeves (T-1-x). In the case of Mobilube 1SHC oil, that difference was insignificant (Figure 6). The nature of permeability variability against time was similar for both oils, i.e., for the standard sleeves (T-1-x), there was quite a rapid decrease in permeability from the value of 10–20 × 10^−15^ m^2^ to the final, stabilized value of about 2 × 10^−15^ m^2^. For the sleeves containing h-BN in the sinter (T-3-x), that decrease was relatively small (for Mobilube 1SHC oil, it was a bit bigger). Overall, it can be said that the permeability characteristics were stable against time (for Hipol, decreases were only 0.2 × 10^−15^ m^2^, and for Mobilube oil, with more than two times lower viscosity at the temperature of 25 °C, the drop was significantly greater, reaching 0.7 × 10^−15^ m^2^).

### 3.2. Load Capacity of PLS-01

The influence of h-BN addition on T-3-x sleeves’ load capacity, in comparison with standard T-1-x sleeves, is shown in Figure 7, Figure 8, Figure 9 and Figure 10. The sleeves were lubricated (impregnated up to about 97–98%) with the following oils: O-3 (Hipol 15F 85 W/90), O-26 (Mobilube 1SHC 75 W/90), and O-30 (mixture 55%/45% of PAO-8 and PAO-40). The results were also related to the results of oils for porous slide bearing lubrication recommended in the literature: Klüberalfa DH3-100 and Antykol TS120 (both lubricants with T-1-x sleeves). As shown in the graphs, for both applied rotational speeds, the values for the recommended lubricants were within the range of the p·v rating, commonly assumed as adequate for the standard porous slide bearings (0.9–2.1 MPa·m·s^−1^). This range has been marked with dashed horizontal lines (Figure 8 and Figure 10).

The presence of h-BN in the structure of the sleeves lubricated with O-3, O-26, and O-30 oils (T-3-x type), caused a significant load capacity increase compared with that of the sleeves not containing boron nitride (T-1-x type). At the rotational speed of *n* = 1400 rpm, over 100% load capacity increase was achieved. Regarding lower rotational speed (*n* = 600 rpm), for O-3 and O-26 oils, an increase of only 25% was registered. The load capacity increase for sleeves lubricated with O-3 or O-26 oils at *n* = 600 rpm would definitely be much higher, but the experiments had to be limited because of the construction and strength restrictions of the test stand (maximum load of 3600 N). In these cases, during all the tests, the sleeves did not seize at all (marked with black arrows).

Achieved values of p_gr_·v rating visibly exceeding 6 MPa·m·s^−1^ were more than three times higher than those of the standard range [17,18,19] required for this type of iron slide bearing (p·v = 0.9–2.1 MPa·m·s^−1^) while ensuring very low resistance to motion (friction coefficient about 0.01) and low operation temperature (below 80 °C) of the bearing.

### 3.3. Load Capacity of PLS-02

The influence of h-BN addition on T-3-x load capacity (linearly increasing load) in comparison with standard T-1-x sleeves is shown in Figure 11 and Figure 12.

The tests with linearly increasing load confirmed the results obtained from using the PLS-01 test stand, presented earlier in the paper. The presence of h-BN in the porous structure of the sleeves lubricated with O-3 or O-26 oil caused a further, significant increase in their load capacity within the entire range of applied speeds, i.e., 600 rpm (v = 0.79 m/s), 1000 rpm (v = 1.31 m/s), and 1400 rpm (v = 1.83 m/s). Due to the permissible load on the PLS-02 test stand (3200 N), it was impossible to seize T-3-x sleeves lubricated with O-3 or O-26 oils, at the rotational speed of *n* = 600 rpm (similar to the tests on the PLS-01 stand), which prevented determination of the actual value of the boundary pressures (and p·v rating). As presented in Figure 11, the load capacity of all sleeves decreased with increasing sliding speed. Such a relationship is, of course, consistent with the current knowledge about these phenomena [8,12,14]. Achieved values of p·v rating, even exceeding 7 MPa·m·s^−1^, were 3.5 times higher than those required for this type of iron slide bearings range, when lubricated with oils recommended by the manufacturers, e.g., Antykol TS120 or Klüberalfa DH3-100.

During the tribological tests on PLS-01 and PLS-02 stands, it was proven that both oils (O-3 and O-26) were correctly performing their function in the self-lubricating porous bearings. However, much better results have been obtained in the case of bearings containing 3% hexagonal boron nitride in the sinter, in comparison to the standard iron bearings. Although the oil permeability in the case of T-3-x sleeves was lower than for standard sleeves, it was stable over time and sufficient to ensure optimum lubrication. Additionally, the effective lubrication was supported by the stabilizing presence of h-BN in the sinter. As a result, the load capacity of the T-3-x bearings was higher than for the standard ones (although the latter have higher permeability). Therefore, the durability of bearings with h-BN in the sinter was also higher, especially in the case of Hipol oil, for which permeability was relatively low. In addition, the amount of lubricating substance lost from the porous structure for Hipol oil was 2–3 times less than for Mobilube oil. As a consequence of a very high (2000 N) applied load (to shorten the test duration time), the durability tests were conducted at a considerably high p_gr_·v rating of 5.24 MPa·m·s^−1^. For the tested oils, Hipol 15F and Mobilube 1SHC, seizure of the bearing was observed after 1100 h and 400 h, respectively. These values are 1.5–2 times higher than for T-1-x bearings working under the same conditions. In comparison with the sleeves impregnated with commonly recommended oils, such as Antykol TS120 or Klüberalfa DH3-100, the mass loss of lubricant is 10 times lower, which results in multiple increases in the durability of T-3-x sleeves lubricated with Hipol oil.

The sleeves containing h-BN in quantities not exceeding 3% were slightly lower in density by about 0.2–0.3 g/cm^3^ and had about 3–4% lower open porosity in comparison with that of traditional ones. Due to lower, but more stable oil permeability of the sinter containing h-BN and some absorptivity of boron nitride caused by its microporosity, a small decrease in iron sinter density and porosity did not worsen the sleeve durability. On the contrary, their durability was somewhat improved, since such bearings can work up to 1.5–2 times longer than the iron bearings without h-BN addition. Moreover, the load capacity of these sleeves increased despite the small decrease in their hardness (about 3 HB), which did not result in higher tribological wear.

## 4. Conclusions

Regardless of the method of research (test stand, conditions), a significant, a twofold increase in the load capacity and durability of the porous sleeves made of iron sinter containing 3% hexagonal boron nitride was achieved. The porous sleeves containing h-BN run quietly and steadily with very low temperature generated by friction, and with a very low friction coefficient. This study demonstrated that the wear of T-3-x sleeves is noticeably lower than the wear of comparatively tested sleeves containing 2.5% of Cu, especially at rotational speeds below 1000 rpm. The addition of 3% of h-BN to the sinter made it possible to receive, at the speed of *n* = 1000 rpm, a p_gr_·v rating of almost 7 MPa·m·s^−1^, i.e., over three times higher than that of a standard range (0.9–2.1 MPa·m·s^−1^) for this type of self-lubricating sliding bearing. Owing to these facts, it is possible to obtain the significant energy savings (electrical, mechanical, etc.) needed for driving mechanisms where porous sleeves are to be used and despite operating under a load several times higher than for bearings of the type currently being used.

## Figures and Tables

**Figure 1 materials-15-05112-f001:**
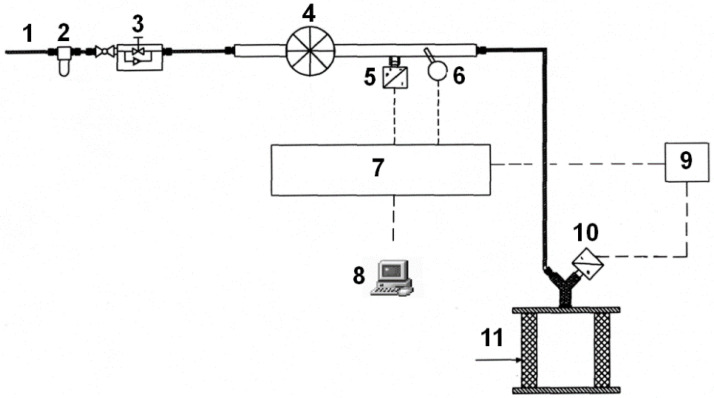
Schematic of the measuring system for assessing permeability: 1—compressor; 2—pressure reducer; 3—throttle-return valve; 4—flowmeter; 5—pressure sensor; 6—temperature sensor; 7—measuring set; 8—PC; 9—measuring set; 10—pressure sensor; 11—tested sample.

**Figure 2 materials-15-05112-f002:**
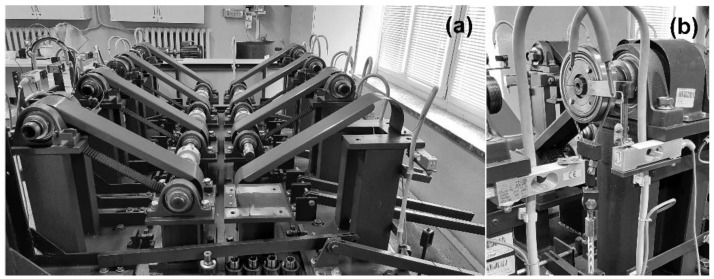
PLS-01 stand (**a**) back view; (**b**) testing shaft, sleeve, and clamp.

**Figure 3 materials-15-05112-f003:**
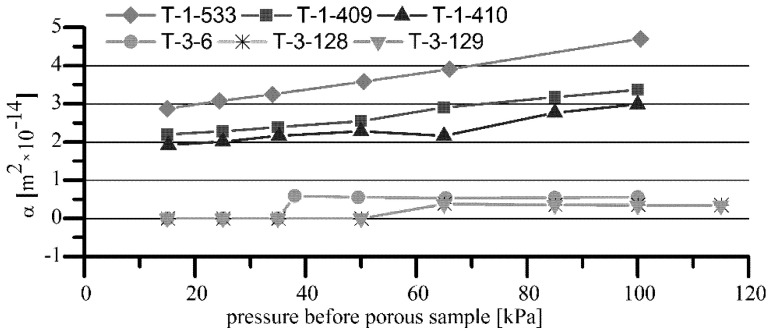
Comparison of T-1-x (standard) and T-3-x (modified) bearings air permeability.

**Figure 4 materials-15-05112-f004:**
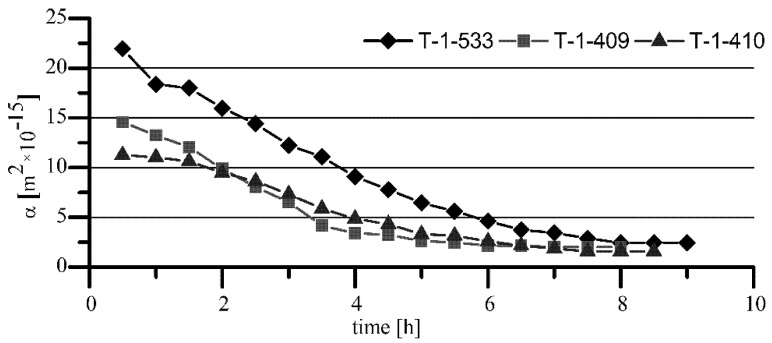
Comparison of T-1-x (standard) and T-3-x (modified) bearings oil permeability.

**Figure 5 materials-15-05112-f005:**
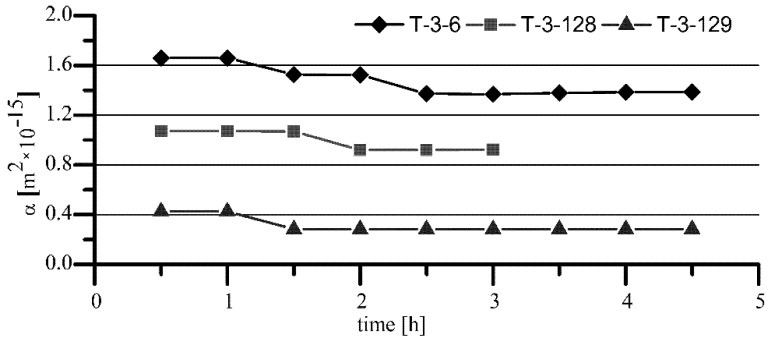
Oil permeability of T-3-x sleeves with addition of BN for Hipol 15F (O-3) oil.

**Figure 6 materials-15-05112-f006:**
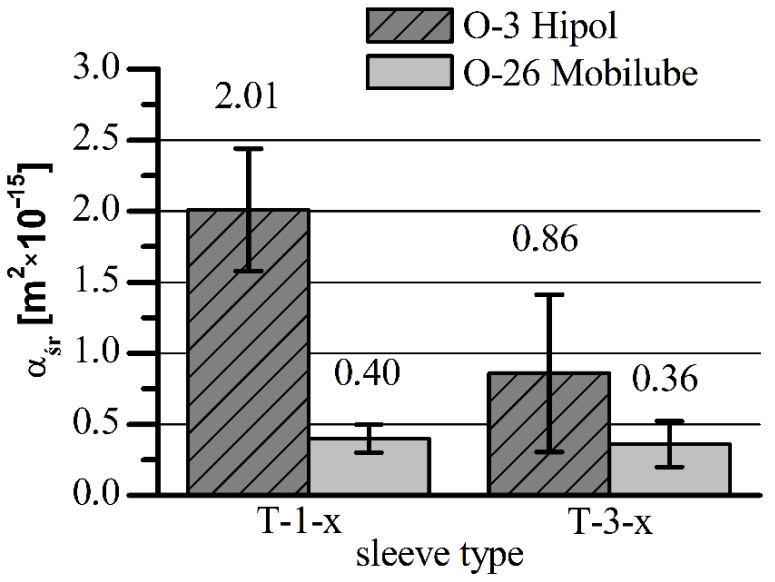
Comparison of the average stabilized values of permeability of the tested sleeves for the two oils: Hipol 15F (O-3) and Mobilube 1SHC (O-26).

**Figure 7 materials-15-05112-f007:**
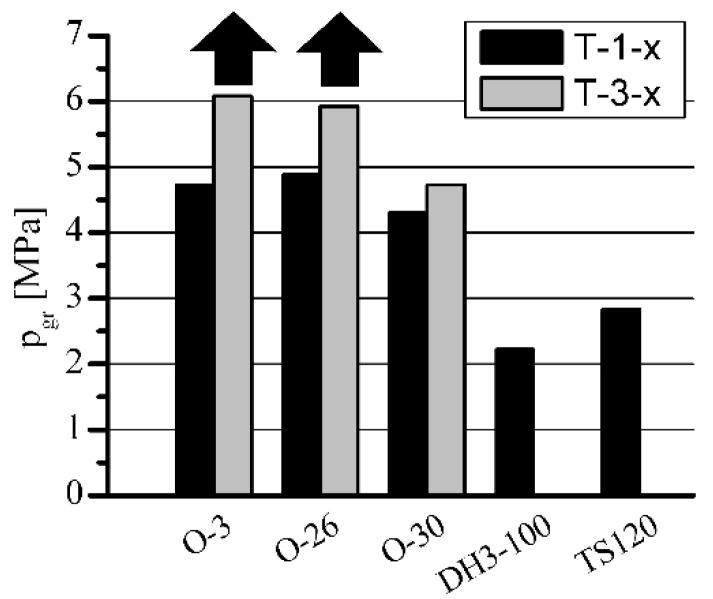
The influence of h-BN on the load capacity of porous sleeves at *n* = 600 rpm.

**Figure 8 materials-15-05112-f008:**
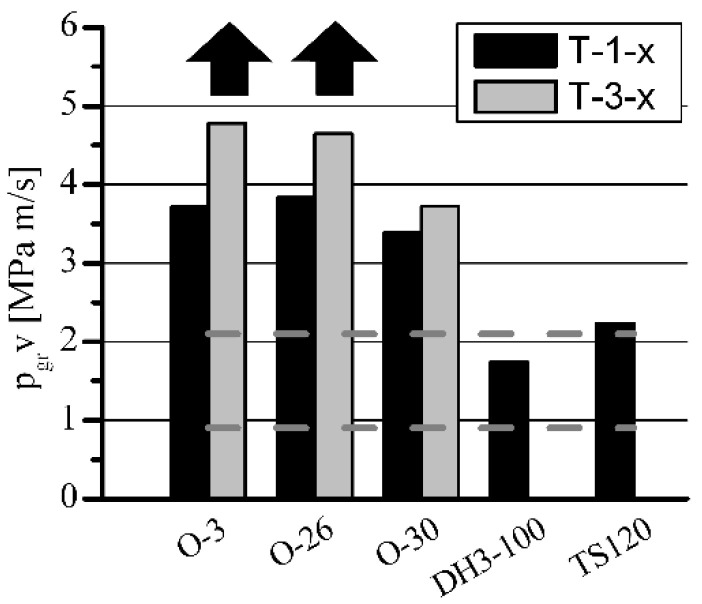
The influence of h-BN on p_gr_·v rating of porous sleeves at *n* = 600 rpm.

**Figure 9 materials-15-05112-f009:**
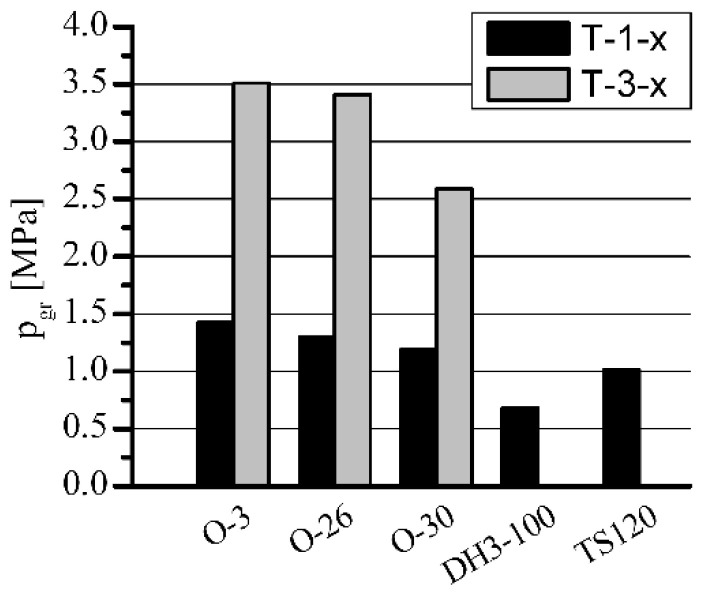
The influence of h-BN on the load capacity of porous sleeves at *n* = 1400 rpm.

**Figure 10 materials-15-05112-f010:**
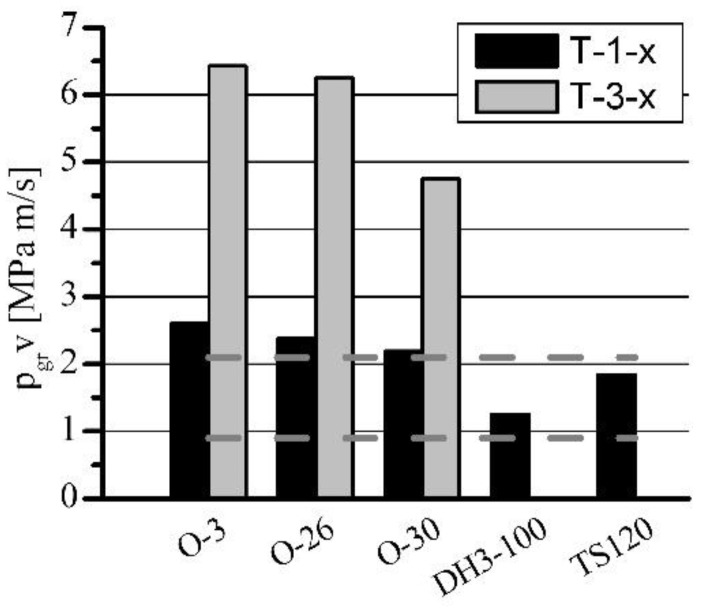
The influence of h-BN on p_gr_·v rating of porous sleeves at *n* = 1400 rpm.

**Figure 11 materials-15-05112-f011:**
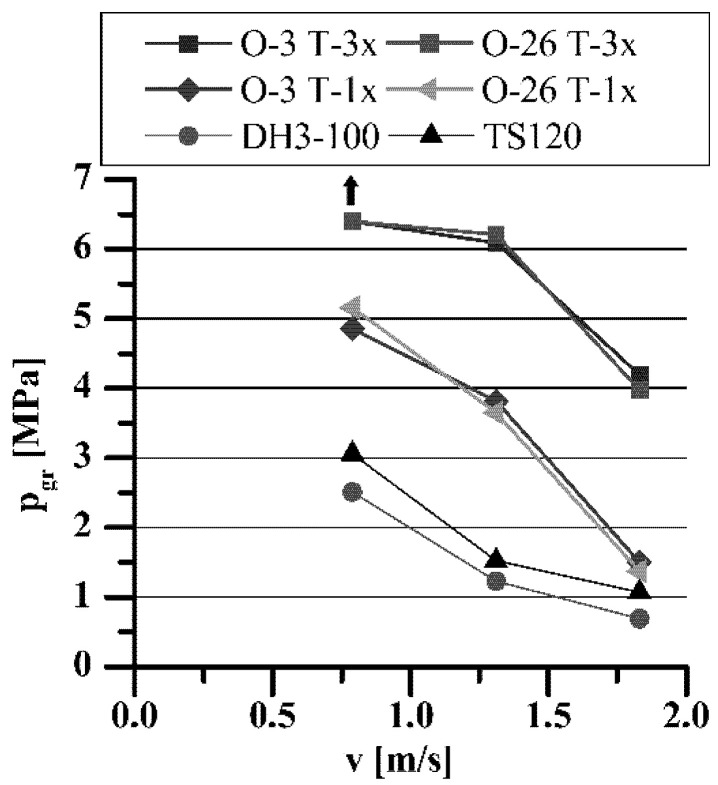
The influence of h-BN on porous sleeve load capacity as a function of sliding speed.

**Figure 12 materials-15-05112-f012:**
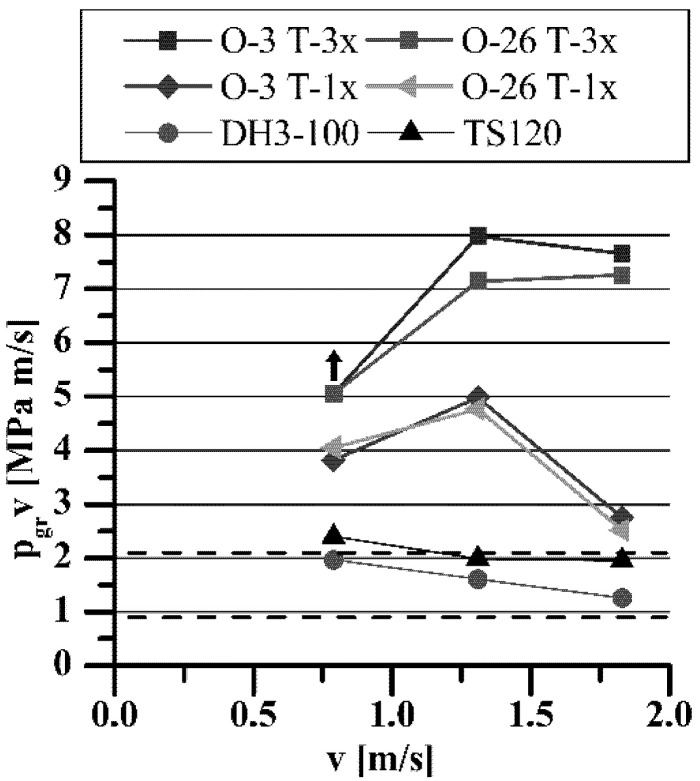
The influence of h-BN on p_gr_·v rating of porous sleeves as a function of sliding speed.

## Data Availability

The data presented in this study are available on request from the corresponding authors.

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
