# Peer review of "Permeability and Load Capacity of Iron Porous Bearings with the Addition of Hexagonal Boron Nitride"

_materials, 2022, doi:10.3390/ma15155112_

Round 1

Reviewer 1 Report

In the paper, authors studied the permeability and bearing capacity of iron-based porous bearings with hexagonal boron nitride. Although the manuscript is about an interesting subject and it could be considered as a reference, contains several aspects that must be improved in order to take into account this work for publication. Major revisions and amount of improvements are necessary.

1.    The abstract is too simple and lacks the expression of research results.

2.    Lack of research background and specific job description in the introduction section.

3.    Please correct the meaning expressed in lines 20, 21, 22 and 25 (e.g. 0.4% ÷1.0%, etc.).

4.    Figure 2 and figure 3 should be combined into one figure (Figure 3 is used as an enlarged view of Figure 2).

5.    The conclusion section is too long and should be summed up succinctly.

Reviewer 2 Report

In this work, authors studied the permeability and load capacity of iron and hexagonal boron nitride with Hipol and Mobilube oil. The article was performed in a thorough scientific data. I have some comments: 

-          Previously, authors studied iron sinters containing 5 wt% of h-BN exhibit a relatively low friction coefficient [ref 10]. Authors should explain why usd 3% of h-BN in this study.

-          It would be useful if authors explain the meaning of “-x” and should give the reason for designing the air infiltration with 25/35 x 20 mm.

-          The details of the test equipment and  chemicals should be added as showed related reference - https://www.mdpi.com/1996-1944/15/6/2052

-          More references should be added for example the advantages of sintered with n-BN.

-          “The tribological properties” should be explained.

Reviewer 3 Report

The article describes the permeability and load capacity of iron porous bearings with additions of BN. Some interesting results are presented in the articles and I believe with some important revisions this article is suitable for publication in the Materials journal. 

1) Overall the English in the article must be improved. In many cases articles such as "the" are missing and a few of the sentences are grammatically incorrect. Please read carefully over the full manuscript and correct where needed. Some examples; the first sentence of the introduction and the first sentence of section 2.2 need attention.

2) The introduction provides some good information about porous iron-based slide bearings but it would be beneficial to add a paragraph discussing the field/applications/significance of their use a bit more. This would provide some nice context for the research.

3) In the experimental section you briefly mention the "preliminary tests" that made you decide on the addition of 3wt% BN. This section would benefit from some more details on this test. The reference provided here is not sufficient to someone who isn't aware of the specific testing. Please add some details of this testing for context.

4) there is no information about how the porous sleeves were produced. Please add a few sentences on the fabrication methods. Also, there is no mention of the way the porosity in these samples was quantified or for example their resulting microstructure. Please add details on these topics. 

5) Section 2.2 - Figures 2 and 3 shod be combined as they show the same piece of equipment. Also, is there a reference for the design of PLS-01?

6) Were all the tests done at room temperature? Please specify. Why was the temperature of 200-220 ˚C (instead of 300 ˚C for example) selected as the seizing criterion?

7) Was there any cracking on the sleeves for the various testing? Was that assessed?

8) Line 192-193, please provide references for this statement. 

9) Line 204 - "As a result", instead of "as the result"

10) Line 212 and various other places throughout the manuscript you use the incorrect symbol. It should be "1.5-2" to signify the range. Please correct throughout. 

11) Line 223 - it should be "noticeably" 

12) Line 232 - please refrain from using "..."

Round 2

Reviewer 1 Report

The revised article has been carefully revised. Although there is still much content for progress in many aspects, including readability, scientificity and innovation, I think it has reached a publishable level, and the overall content of the article still needs to be strengthened.